# Nuclei Segmentation in Histopathological Images with Enhanced U-Net3+

**Bishal Ranjan Swain**[1]                BISHALSWAIN@KUMOH.AC.KR
**Kyung Joo Cheoi** [2]                KJCHEOI@CHUNGBUK.AC.KR
**Jaepil Ko** [1,*]                NONEZERO@KUMOH.AC.KR

[1] *Dept. of Computer Eng., Kumoh National Institute of Technology, Gumi, Korea*

[2] *Dept. of Computer Science, Chungbuk National University, Cheongju, Korea*

[*] *Author to whom correspondence should be addressed.*

**Editors:** Accepted for publication at MIDL 2024

## Abstract

In the rapidly evolving field of nuclei segmentation, there is an increasing trend towards developing a universal segmentation model capable of delivering top-tier results across diverse datasets. While achieving this is the ultimate goal, we argue that such a model should also outperform dataset-specific specialized models. To this end, we propose a task-specific feature sensitive U-Net model, that sets a baseline standard in segmentation of nuclei in histopathological images. We meticulously select and optimize the underlying U-Net3+ model, using adaptive feature selection to capture both short- and long-range dependencies. Max blur pooling is included to achieve scale and position invariance, while DropBlock is utilized to mitigate overfitting by selectively obscuring feature map regions. Additionally, a Guided Filter Block is employed to delineate fine-grained details in nuclei structures. Furthermore, we apply various data augmentation techniques, along with stain normalization, to reduce inconsistencies and thus resulting in significantly outperforming the state-of-the-art performance and paving the way for precise nuclear segmentation essential for cancer diagnosis and possible treatment strategies.

**Keywords:** Nuclei segmentation, Histopathological images, Segmentation

## 1. Introduction

Nuclei Segmentation of histopathological images play an important role in medical diagnostics of Hematoxylin and Eosin (H&E) stained tissues as it aids pathologists in understanding morphology of cellular structures (Clayton, 1991; Elston and Ellis, 1991). This not only helps in cancer grading but also in predicting the effectiveness of various treatments (Kumar et al., 2017). Finding and segmenting individual nuclei in histopathological images is one of the crucial steps involved in numerous analytical procedures of cancer diagnosis (Cui et al., 2019; Imtiaz et al., 2023). However, segmentation of nuclei in histopathological images is challenging as there are large variations in color, texture, and shape. Moreover, the variability in staining procedures used during the image procurement processes further increase inconsistencies in the images (Sampias and Rolls).

Recent advances have seen the rise of convolutional neural network (CNN)-based deep learning models that have started to gain attention due to their robust and consistent performance in semantic segmentation (Zhang et al., 2020). Among these, U-Net (Ronneberger et al., 2015) has emerged as the most influential architecture, introducing encoder-decoder

structure that has been highly effective for medical image segmentation tasks (Liu et al., 2023). It is a fully convolutional network (FCNN) that captures details through deep layers and upscales images, merging simple and complex features using skip connections for precise segmentation (Li et al., 2023b). The introduction of U-Net++ (Zhou et al., 2018) aimed to improve upon U-Net by implementing nested and dense skip pathways, enhancing feature propagation and reducing the semantic gap between encoder and decoder layers. MBU-TransNet (Qiao et al., 2023) introduced a fusion of multi-branch U-shaped networks with transformer architecture to leverage long-range dependencies. CFNet (Zhan et al., 2023) proposed a cross-scale feature fusion method to enhance feature selection across different scales, but it may not fully address the challenge of capturing highly detailed local features alongside global context. U-Net3+ (Huang et al., 2020) was introduced with full-scale connections and deep supervisions that sought to capture multi-scale features more effectively. However, variations in nuclear morphology, staining quality, and image resolution inherent in different datasets can notably impact the performance of U-Net based models (Li et al., 2023a).

Data augmentation plays a crucial role in generation of data required for proper training of a deep learning model. Among many data augmentation techniques, adding Gaussian noise has been a common choice to simulate random perturbations and variation in images (Wyatt et al., 2022). However, Gaussian noise adds perturbations that are statistically uniform across the image and lacks spatial coherence that are often present in real-world artifacts and variations (Bae et al., 2018). Moreover, the random fluctuations introduced by Gaussian noise do not adequately represent the artifacts or noise variations in medical images (Kascenas et al., 2023). To overcome the shortages of human-labeled data we implement Perlin-noise based data augmentation strategy on pathological images. Perlin noise provides random but natural looking patches or textures, with little computational cost (Perlin, 1985).

Recently, there has been a rise in the concept of universal or foundational models in segmentation, aiming to develop adaptable and dataset-agnostic models. These models can be trained once and then applied to a wide range of segmentation tasks (Ma et al., 2023). The Segment Anything Model (SAM) (Kirillov et al., 2023), in particular has been trained with a massive dataset of mask labels, making it highly adaptable to a wide range of tasks and there have been various other models like AutoSAM (Shaharabany et al., 2023) that utilize SAM architecture in making more robust and models with high performance. However, this broad adaptability of such models can lead to compromised performance in highly specialized tasks. Study showed that SAM based model do not consistently achieve satisfying performance for dense segmentation tasks like in pathology images (Deng et al., 2023). Moreover we also performed initial test experiments on generalized U-Net based model like nnUNet (Isensee et al., 2018), but it did not perform better than U-Net3+ as shown in the ablation studies. To this end, our study proposes a nuclei segmentation-specific enhanced U-Net3+ model that captures the nuclear morphology and staining variations to surpass the state-of-the-art segmentation results. Inspired from the skip connections of U-Net3+ (Huang et al., 2020), we implement our model to simultaneously capture both local and global representations, introduce Gated Linear Units (GLU) (Dauphin et al., 2017) within the convolution layers for adaptive feature selection and thereby allowing the model to selectively focus on both local and global features effectively, incorporate max blur

pooling (Zhang, 2019) to solve scale and position invariance and anti-aliasing problems that are inherently found in encoder-decoder architectures, utilize DropBlock (Ghiasi et al., 2018) to mitigate overfitting by selectively obscuring feature map regions and using a Guided Filter Block (Wu et al., 2019) to delineate fine-grained details in nuclei structures. Furthermore, we perform several pre-processing on the data including stain normalization to account for staining inconsistencies and data augmentation techniques that are meticulously tailored to address the inherent variability in the pathological H&E stained images. Through this specialized approach, our model outperforms the current state-of-the-art models by some margin in nuclei segmentation of histopathological images.

## 2. Materials and Methods

### 2.1. Dataset

This study utilizes the MoNuSeg2018 dataset (Kumar et al., 2017) for training and evaluating our model. The dataset includes 30 training images and 14 test images, encompassing a total of 21,623 manually annotated nuclei for training and 7,223 for testing. These images were derived from H&E-stained whole slide images, featuring tissues from breast, kidney, liver, prostate, bladder, colon, and stomach in the training set, and an additional inclusion of lung and brain tissues in the testing set. All images were acquired at a 40x magnification, offering high-resolution insights into the cellular structures. In addition to MoNuSeg2018, the study further uses the CPM-17 (Vu et al., 2018) and CoNSEP (Graham et al., 2019) datasets for comparative and comprehensive evaluation.

### 2.2. Data Preprocessing

#### 2.2.1. Stain Normalization

Stain normalization addresses the variability in histopathological images due to differences in H&E staining procedures, which affects the appearance of cellular structures. Variations in hues and intensities arise from disparate staining protocols, lighting, and imaging equipment, posing challenges for models trained on specific staining conditions. To mitigate these issues, we adopted a computationally efficient stain normalization approach, following the method described by (Mahbod et al., 2019). A reference image from the training set was selected for Reinhard normalization, aligning all dataset images to this standard (Reinhard et al., 2001). Reinhard normalization was chosen for its proven effectiveness in previous related researches (Patil et al., 2021).

#### 2.2.2. Data Augmentation

Histopathology images inherently contain a high degree of variability, not only in the stains but also in the positions and morphological characteristics of cellular structures. Keeping in consideration, various photometric and geometric augmentations were performed to handle limited number of samples in the datasets.

- Geometric and Photometric Augmentations: Performing augmentations that alter the angular positions and shapes of the nuclei can differ the training samples significantly.

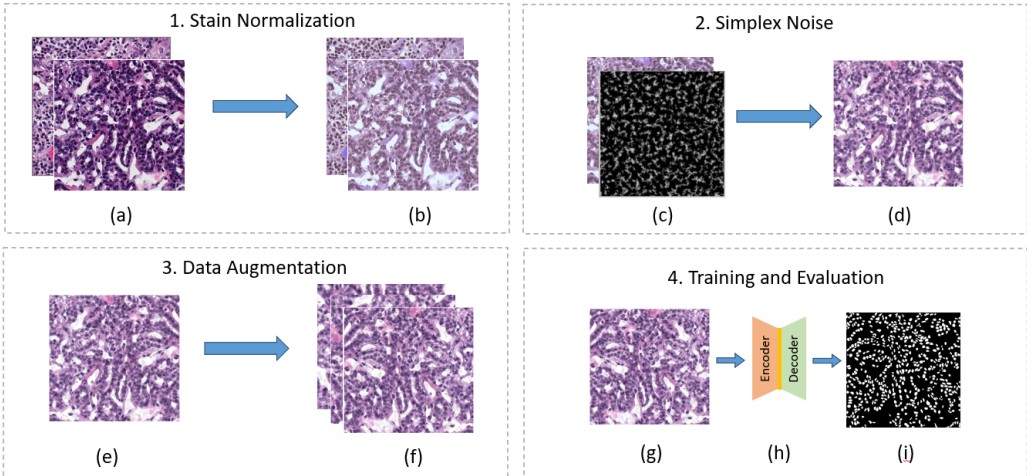

Figure 1: The flowchart of the proposed methodology. 1. Stain normalization: (a) original images, (b) stain normalized images; 2. Simplex noise: (c) simplex noise and stain image, (d) noise added image; 3. Image augmentation: (e) noise added image, (f) augmented images; 4. Training and Evaluation: (g) pre-processed image, (h) proposed model, (i) predicted segmentation mask.

Basic geometric augmentations such as rotations, scaling, flips were performed to obtain samples from different perspectives and scales. Additionally, elastic deformations that mimic the natural deformations in biological tissues were performed that helped in enhancing the model's ability to handle non-rigid transformations and complex variations. Photometric augmentations like Gamma and intensity level transformations along with contrast limited adaptive histogram equalization were performed.

- Perlin Simplex Noise: The Perlin noise algorithm is employed to generate a pseudo-random gradient vector on each corner on a given grid. Next, it calculates the distance vectors from a given position to the surrounding corners on the grid. Then, it takes the dot product between the distance vector and the gradient vector, thereby obtaining influence values. The dot product becomes positive if the two vectors are pointing in the same direction, and it becomes negative if the two vectors are pointing in opposite directions. In the final step, interpolation is performed between these influence values to construct smooth patterns within the grid. In our experiments, we generated 2D Perlin noise with the same size of the image patches and concatenated them to produce noisy images.

## 2.3. Proposed Methodology

### 2.3.1. Model Overview

We implement a modified and refined U-Net3+ model for nuclei segmentation inspired from the full-scale skip connection implementation. We selected U-Net3+ as the baseline as it

performed better on our initial experiments as detailed in Appendix. Key adjustments to U-Net3+ (Huang et al., 2020) include reducing the model's depth to mitigate overfitting on small datasets, optimizing feature channels to capture essential details without excessive complexity and using GLU for gated feature selection.

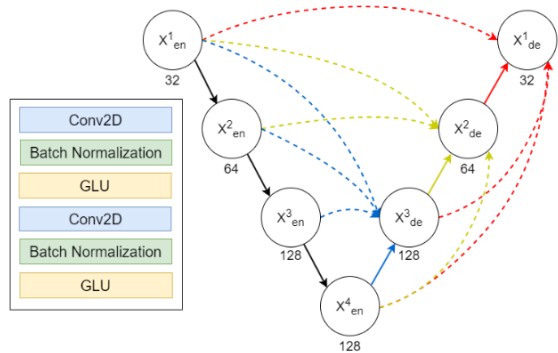

Figure 2: Structural overview of the proposed enhanced U-Net3+ model.

- Optimization of Network Capacity: To prevent the model from memorizing rather than learning, we reduced the depth of U-Net3+ by eliminating one layer. Reducing the capacity of the model compels it to focus on the most important features such as – cell boundaries and tissue types and prevents it from focusing on irrelevant details.

- Appropriate Feature Channels: Feature channels help in capturing specific characteristics of the input, such as edges, corners, or textures. The initial layer of the encoder captures the low-level features like edges and corners and the deeper layers capture high-level abstract features. The specific arrangement of 32, 64, 128 and 128 was chosen as it empirically performed better than other configurations as in Table 4.

### 2.3.2. ADAPTIVE FEATURE SELECTION

The importance of features can vary significantly across different regions in a histopathological image. Certain areas may contain more nucleus and might require more nuanced feature extraction, while others may be relatively homogeneous like cytoplasm and might require less detail. While traditional activation functions like ReLU (Agarap, 2019), apply the same transformation across all features, GLUs can learn to 'gate' certain features selectively. A GLU takes its input and splits it into two halves. One half is transformed linearly (much like a standard unit), and the other is transformed via a sigmoid activation function. The output of the sigmoid is used as a gate to control the information flow from the linear half. Mathematically, for an input $x$,

$$GLU(x) = sigmoid(W_a \cdot x + b_a) \odot (W_i \cdot x + b_l) \tag{1}$$

Here, $\odot$ denotes the element wise multiplication, $W_a, b_a$ are the weights and biases for the gated mechanism, and $W_l, b_l$ are the weights and biases for the linear transformation.

### 2.3.3. Further Refinements

- Shift Invariant Convolutions: To address aliasing artifacts from max pooling, we implemented max blur pooling. This approach, involving a blurring step post-pooling, smooths feature maps to reduce aliasing and enhance the model's ability to capture fine details.

- DropBlock Regularization: DropBlock is a structured form of dropout that removes contiguous regions from the feature maps. By enforcing the model to fill in missing regions based on surrounding context, DropBlock encourages the network to learn more robust and generalized feature representations.

- Trainable Guided Filter: The trainable guided filter (Wu et al., 2019) offers a sophisticated mechanism for refining segmentation masks by leveraging edge-preserving filtering. It takes the coarse segmentation map produced by the network and refines it using the original input image as a guide. The guided filter performs smoothing within regions while preserving edges, effectively capturing both the global structure and local details of the targeted nuclei without introducing the halo artifacts.

### 2.3.4. Combined Loss Function

The model is trained using combined loss function by integrating the weighted average dice loss and focal loss (Lin et al., 2018). Weighted dice loss is commonly used in biomedical imaging applications where there is infrequency in occurrence of certain regions or features. The focal loss on the other hand focuses more on the hard negatives and down-weights the easy examples, thereby allowing the model to focus on more challenging regions. $L_{seg}$ is the loss used in the experiments where, $\alpha$ and $\beta$ are the controlling parameters of the respective loss components.

$$L_{seg} = \alpha L_{dice}(y, p(y)) + \beta L_{focal}(y, p(y)) \tag{2}$$

## 3. Experiments and Results

The experiments were conducted on NVIDIA A6000 GPU, Intel i7 6700 CPU, running on the Ubuntu 22.10 operating system and using PyTorch framework. The images in training set was split into training and validation sets with ratio of 8:2 and then were augmented. The batch size was set to be 16 and number of epochs to be 50. Several experiments were conducted and through empirical results the optimal learning rate was found to be 1e-6. For the performance evaluation of the model, widely used evaluation criterion of Intersection-over-Union (IoU) (Rezatofighi et al., 2019) and Dice-Score (Eelbode et al., 2020) were used across experiments. The Dice Score computes the overlap between the predicted segmentation mask (A) and the ground truth mask (B). The IoU metric evaluates the quality of the object-level segmentation by calculating the overlap between the predicted and ground truth masks for each class and then averaging these overlaps.

Our experiments demonstrate a significant improvement in nuclei segmentation performance across various datasets, as evidenced by Table 1 and Table 2. Our optimized model,

showcases superior results compared to previous and present existing state-of-the-art methods, including U-Net, U-Net++, CFNet, MBUTransNet, nnUNet, SAM, and AutoSAM. Specifically, on the MoNuSeg test images, our model achieves the highest Dice coefficient of 0.8902 and mIoU of 0.7924, outperforming other methods. This performance is not isolated to the MoNuSeg dataset but extends across CPM-17 and CoNSep datasets, with our model outperforming other approaches with Dice scores of 0.9325 and 0.8172, and mIoU scores of 0.8776 and 0.7257, respectively. The model was trained individually across the datasets.

Table 1: Comparison on results obtained from MoNuSeg test images using previous methods. Underlined values denote the baseline results.

| Method | Dice | mIoU |
|---|---|---|
| U-Net (Ronneberger et al., 2015) | 0.7943 | 0.6599 |
| U-Net++ (Zhou et al., 2018) | 0.7949 | 0.6604 |
| CFNet (Zhan et al., 2023) | 0.7987 | 0.6668 |
| MBUTransNet (Qiao et al., 2023) | 0.8160 | 0.6902 |
| U-Net3+ (Huang et al., 2020) | 0.8260 | 0.7039 |
| nnUNet (Isensee et al., 2018) | 0.8031 | 0.6781 |
| SAM (Kirillov et al., 2023) | 0.6950 | 0.6187 |
| AutoSAM (Shaharabany et al., 2023) | 0.8242 | 0.7017 |
| **Ours** | **0.8902** | **0.7924** |

Table 2: Comparison of proposed model performances across datasets. Underlined values denote the baseline results.

| Models | MoNuSeg | | CPM-17 | | CoNSep | |
|---|---|---|---|---|---|---|
| | Dice | mIoU | Dice | mIoU | Dice | mIoU |
| U-Net | 0.7943 | 0.6599 | 0.8312 | 0.7759 | 0.7192 | 0.6260 |
| U-Net++ | 0.7949 | 0.6604 | 0.8471 | 0.7891 | 0.7416 | 0.6433 |
| UNet3+ | 0.8260 | 0.7039 | 0.8619 | 0.8042 | 0.7784 | 0.6829 |
| **Ours** | **0.8902** | **0.7924** | **0.9325** | **0.8776** | **0.8172** | **0.7257** |

The results underscore the effectiveness of our approach, which incorporates techniques such as stain normalization, simplex noise, GLU, max blur pooling, dropblock and trainable guided filter. These enhancements contribute to the model's ability to more accurately distinguish between nuclei and surrounding tissue, even in the presence of staining variability and complex tissue morphology. Table 1 and Table 2 draw the comparative quantitative analysis while Figure 3 draws the qualitative superiority of our approach. We also performed statistical analysis, using a two-sample t-test (Welch's t-test for unequal variances), got a p-value of approximately 0.0134. This result of p-value $< 0.05$ suggests that the performance difference observed between your model and the base model is unlikely to have occurred by chance and that our model has a statistically significant improvement (Fu et al., 2024).

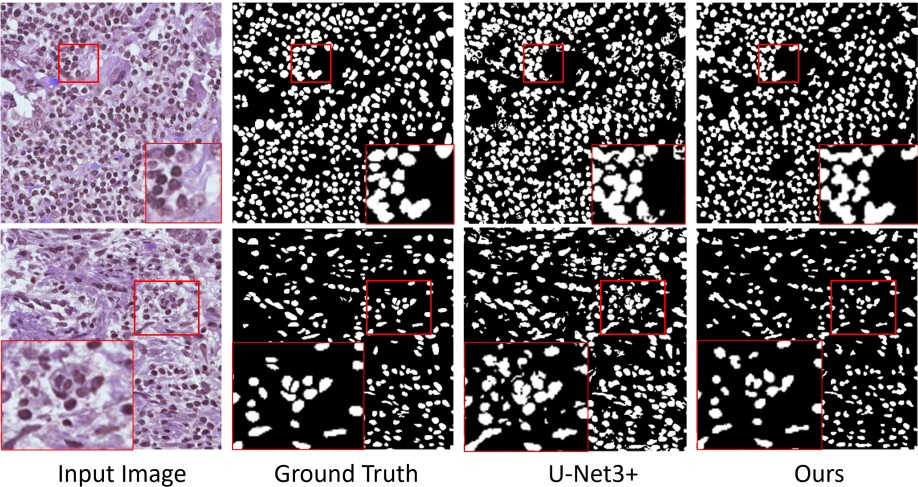

Input Image      Ground Truth      U-Net3+      Ours

Figure 3: Comparison of segmentation results of our proposed model on MoNuSeg dataset with U-Net3+.

## 4. Conclusion

In this study, we introduced a nuclei segmentation-specific enhanced U-Net3+ model in H&E stained histopathology images. Our implementation included - enhancing the performance of U-Net3+ model by implementing a series of optimizations including adaptive feature selection through Gated Linear Units (GLUs), max blur pooling for scale and position invariance, DropBlock regularization to mitigate overfitting and fast trainable guided filter for efficient learning. Furthermore, we applied stain normalization to achieve consistency across images and utilized advanced data augmentation techniques to expand the training dataset. Our methodology significantly outperformed previous methods including the existing state-of-the-art models. This was achieved by addressing key challenges such as data imbalance with a hybrid combined loss function that enhanced the model's sensitivity to varying object sizes and class imbalances. Despite the promising results, the segmentation of overlapping and clumped nuclei in H&E stained images remains a challenge. Future work will focus on developing a more intricate model architecture to capture that takes advantage of the foundational and task-specific models using instance segmentation. Additionally, we plan to explore the potential of transformers and promptable segmentations for further advancements in nuclei segmentation. Our study demonstrates that strategic optimizations can lead to significant improvements in histopathological image analysis, thereby laying a critical groundwork for more accurate cancer diagnosis and informing potential treatment pathways.

## Acknowledgments

This research was supported by BL Science grants (202301630001).

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

## Appendix A. Ablation Studies

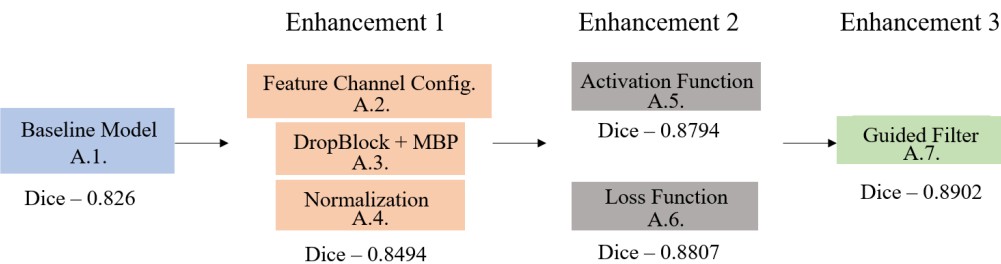

Figure 4: An overview of the ablation study which details the enhancements performed on the base model.

This section meticulously evaluates the incremental contribution of each model design choice as proposed enhancement to the baseline. Sub-section A.1., details the experiments

conducted for the selection of baseline and similarly section A.2., A.3., A.4., detail about the effectiveness of feature channel configuration, dropblock + max blur pooling, and normalization respectively. A.5. and A.6., detail the experiments and verify the effectiveness of the activation and loss functions implemented. Finally A.7. shows the effectiveness of the fast trainable guided filter in the model.

### A.1. Baseline Model Performance

The baseline performance comparison between U-Net3+, ELU-Net (Deng et al., 2022) and nnUNet (Isensee et al., 2018) on MoNuSeg test images establishes U-Net3+ as superior, with an average Dice score of 0.826 compared to ELU-Net's 0.808 and nnUNet's 0.796. Table 3 illustrates the performance of the models across all the test image cases. This highlighted U-Net3+'s efficiency in capturing nuanced features across various tissue types compared to ELU-Net and nnUNet (Isensee et al., 2018), setting a robust foundation as baseline for further enhancements.

Table 3: Performance in terms of Dice scores of MoNuSeg test images on U-Net3+, ELU-Net and nnUNet for creating baseline

| Model | 0 | 1 | 2 | 3 | 4 | 5 | 6 | 7 | 8 | 9 |
|---|---|---|---|---|---|---|---|---|---|---|
| U-Net3+ | 0.825 | 0.905 | 0.885 | 0.917 | 0.751 | 0.786 | 0.786 | 0.886 | 0.887 | 0.849 |
| ELU-Net | 0.804 | 0.896 | 0.881 | 0.898 | 0.712 | 0.777 | 0.79 | 0.868 | 0.874 | 0.811 |
| nnUNet | 0.796 | 0.900 | 0.891 | 0.892 | 0.686 | 0.782 | 0.807 | 0.863 | 0.874 | 0.787 |

| 10 | 11 | 12 | 13 | AVG |
|---|---|---|---|---|
| 0.904 | 0.578 | 0.839 | 0.758 | **0.826** |
| 0.859 | 0.51 | 0.842 | 0.784 | **0.808** |
| 0.828 | 0.455 | 0.858 | 0.823 | **0.803** |

### A.2. Feature Channel Configurations

Table 4 evaluated the performance of various feature channel configurations in the U-Net3+ model adapted for MoNuSeg data. The table compares the parameters ('#params') and the dice coefficient ('dice') for each configuration. The configuration [32, 64, 128, 128] was found to be most effective which means that the first layer has 32 feature channels, second layer has 64, third layer has 128 and the fourth layer has 128. The optimal configuration demonstrates that a strategic increase in channel depth at later stages can significantly enhance segmentation accuracy without excessively inflating the model's parameter count. This suggests an effective method for maximizing performance while maintaining computational efficiency.

Table 4: Model performance on different configurations of feature channel combinations

| Configuration | #params | Dice |
|---|---|---|
| 16, 32, 64, 128 | 1.02M | 0.8483 |
| 32, 32, 64, 128 | 1.26M | 0.8372 |
| 32, 64, 64, 128 | 1.34M | 0.8443 |
| 32, 32, 128, 128 | 1.50M | 0.8493 |
| 32, 64, 128, 128 | 1.60M | 0.8494 |
| 32, 64, 128, 256 | 2.2M | 0.8239 |

## A.3. DropBlock and Max Blur Pooling Implementations

The comparison of dropout strategies underscores DropBlock's effectiveness over traditional dropout in spatially structured data like histopathological images. Integrating DropBlock, along with Max Blur Pooling (MBP), leads to the highest Dice and mIoU scores in Table 5, indicating its pivotal role in enhancing model generalization and mitigating overfitting by encouraging spatially distributed feature learning.

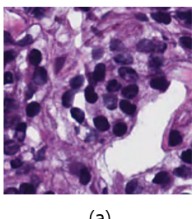
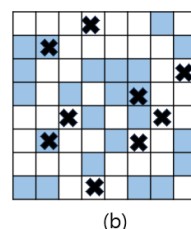
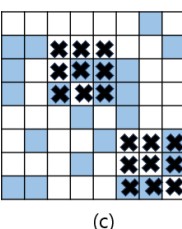

(a)  (b)  (c)

Figure 5: Visualization of DropBlock. (a) Input image to the network. The blue regions in (b) and (c) include the activation units which contain semantic information in the input image. Dropping out activations at random is not effective in removing semantic information because nearby activations contain closely related information. Instead, dropping continuous regions can remove semantic information (that is the whole nucleus or area surrounding nucleus) and consequently enforcing remaining units to learn features for classifying input image.

## A.4. Stain Normalization and its Need

The analysis of stain normalization techniques underscores the critical role of selecting an appropriate target image for both macenko (Macenko et al., 2009) and reinhard normalization as shown in Figure 6. The target image was chosen after meticulous evaluation of the dataset to ensure compatibility with the majority of images. Figure 7 illustrates the process, showing the original, target, and Reinhard-normalized images. This approach emphasizes the importance of a careful selection process to maintain consistency across

Table 5: Model performance on different dropout strategies

| Feature | Dice | mIoU |
|---|---|---|
| None | 0.8188 | 0.6946 |
| Dropout | 0.8361 | 0.7164 |
| DropBlock | 0.8488 | 0.7246 |
| Dropout+MBP | 0.8379 | 0.7168 |
| DropBlock+MBP | 0.8494 | 0.7251 |

the dataset. The effectiveness of this normalization technique is quantitatively validated in Table 6, demonstrating the effectiveness of Reinhard normalization and highlighting its suitability for preserving histological details essential for accurate segmentation.

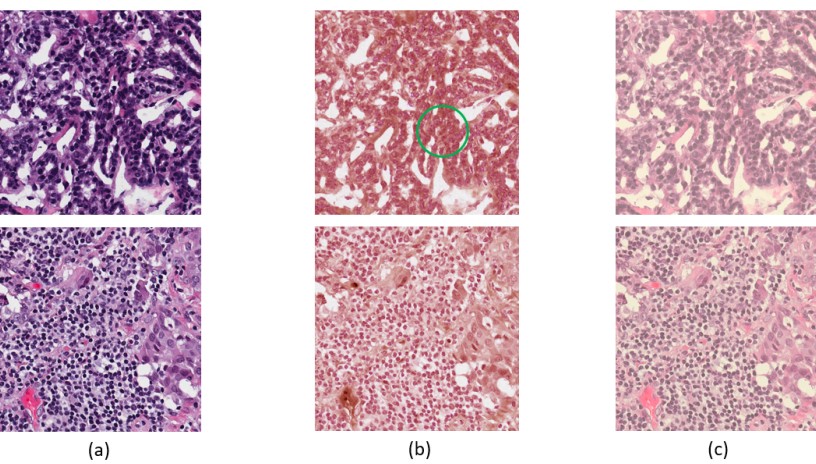

(a)  (b)  (c)

Figure 6: Visualization of normalized images and need of selecting correct target image. (a) the original image, (b) Macenko normalized image, (c) Reinhard normalized image. The green circle highlights the ambiguous region in the image where, the normalization can make it more difficult to segment the images. Therefore there is a need to correctly select the target image for normalization.

Table 6: Model performance on stain normalization

| Feature | Dice | mIoU |
|---|---|---|
| None | 0.8263 | 0.7059 |
| Macenko | 0.8361 | 0.7164 |
| Reinhard | 0.8494 | 0.7251 |

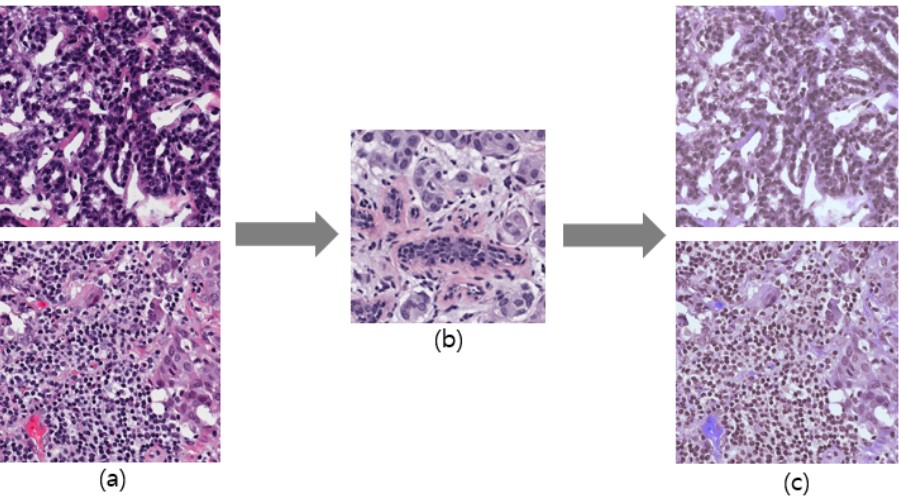

Figure 7: Stain Normalization. (a) Unnormalized images, (b) target image, (c) reinhard normalized images

### A.5. Activation Function Selection

We perform several experiments to compare the effectiveness of GLU activation compared to ReLU, Leaky ReLU, and Swish in terms of Dice and mIoU scores. Superior performance of GLU shows its ability to adaptively filter and propagate relevant features through the network underlines its utility in complex segmentation tasks, suggesting a promising direction for enhancing model sensitivity and specificity.

Table 7: Model performance comparison with ReLU and GLU activation functions

| Activation Function | Dice | mIoU |
|---|---|---|
| ReLU | 0.8488 | 0.7246 |
| Leaky ReLU | 0.8494 | 0.7251 |
| Swish | 0.8486 | 0.7244 |
| GLU | 0.8794 | 0.7879 |

The sigmoid gating mechanism in GLU can learn to turn features on and off adaptively to learn the features that are important and focus on them. It makes the feature maps sparse by not focusing on less important features (Liu et al., 2022). In the encoder-decoder like architecture in U-Net3+, the earlier layers capture local features and the deeper layer capture global features. GLUs are placed across multiple layers to adaptively learn to gate features at different scales and balance the need between local and global details. It

helps the model learn local features like edges of the nuclei, texture or tissue segment and long-range global dependencies like shape and arrangement of cluster of nuclei.

## A.6. Loss Function Comparisons

In the assessment of various loss functions including - BCE, Dice, Focal and Combined Focal and Dice illustrates the superior performance of the combined Focal and Dice loss, achieving the highest Dice and mIoU scores. This combination effectively balances the model's attention between prevalent and rare segmentation targets, optimizing the learning process towards challenging regions and improving overall segmentation accuracy.

Table 8: Model performance on various loss functions

| Loss Functions | Dice | mIoU |
|---|---|---|
| BCE | 0.8436 | 0.7243 |
| Dice | 0.8541 | 0.7289 |
| Focal | 0.8694 | 0.7779 |
| Focal + Dice | 0.8807 | 0.7897 |

## A.7. Fast Trainable Guided Filter Block

The usage of a fast trainable guided filter from (Wu et al., 2019) was implemented at the end of the network before the sigmoid layer. As indicated in Table 9, the inclusion of the guided filter showed an increase in both dice and mIoU scores. Visual representation of with and without fast guided filter is shown in Fig. 8.

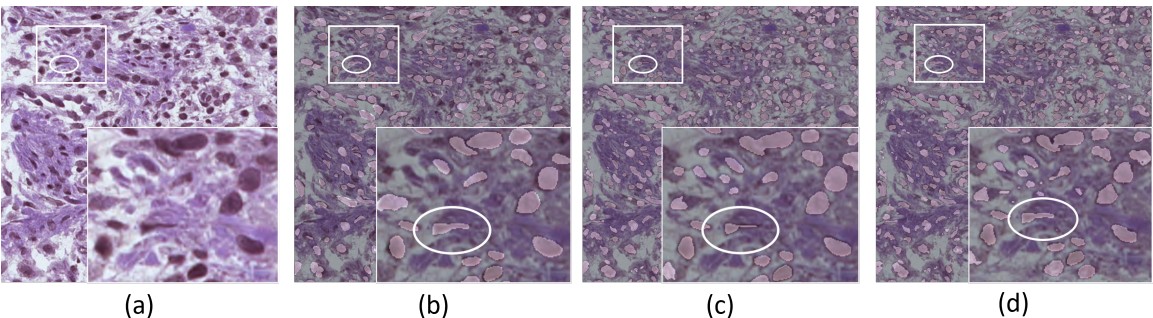

Figure 8: Visualization for the effects of guided filter. (a) input image, (b) overlayed target image, (c) predicted image before guided filter and (d) predicted image after using guided filter.

Table 9: Model performance on with and without guided filter

| Feature | Dice | mIoU |
|---|---|---|
| w/o Guided Filter | 88.07 | 78.97 |
| w/ Guided Filter | 89.02 | 79.24 |

