# OpenReview forum: "Nuclei Segmentation in Histopathological Images with  Enhanced U-Net3+"
_MIDL.io/2024/Conference — MIDL 2024 Poster_

### Official Review · Reviewer_rzgC · 2024-02-26

**Confidence:** 4
**Preliminary Rating:** 3
**Final Rating:** 3.5

**Summary:**

This study introduces an enhanced U-Net3+ model for the semantic segmentation of nuclei in histopathological images, aiming to establish a new baseline in the field. By integrating adaptive feature selection, max blur pooling, DropBlock, and a Guided Filter Block, the model seeks to capture both short- and long-range dependencies, achieve scale and position invariance, mitigate overfitting, and delineate fine-grained details in nuclei structures. Additionally, data augmentation techniques and stain normalization are applied to reduce inconsistencies across datasets. The authors state that their approach significantly outperforms current state-of-the-art models, thereby contributing to the precision of nuclear segmentation crucial for cancer diagnosis and treatment strategies.

**Strengths:**

- The paper's focus on enhancing the U-Net3+ architecture through various innovative techniques represents an interesting contribution to the field of nuclei segmentation in histopathological images.
- The comprehensive approach, combining adaptive feature selection, max blur pooling, DropBlock, and a Guided Filter Block, is a robust attempt to address the common challenges in nuclei segmentation, such as scale variance and overfitting.
- The application of data augmentation and stain normalization techniques to improve model generalization across different datasets showcases the authors' attention to the practical challenges in histopathological image analysis.
- The manuscript is well-structured, providing a clear and concise presentation of the proposed method, experimental setup, and findings.

**Weaknesses:**

- The paper does not discuss the potential of nnUnet [1], which is a notable omission given its performance parity with or superiority to U-Net++ in similar tasks.
- The lack of consideration for instance segmentation approaches, particularly in the context of overlapping and clumped nuclei, is a missed opportunity to address a critical challenge in the field. Although mentioned in the discussion, this is crucial for real application.
- Evaluating the performance on different cell complexity levels (e.g. difficulty levels in terms of annotation) would be very interesting to see e.g. using the data set provided in [2].
- The paper's claims of significant performance improvements are not substantiated with statistical proof, weakening the strength of these assertions.
- A combined ablation study to quantify the contribution of each proposed enhancement to the overall performance improvement is missing, which would provide valuable insights into the efficacy of the individual components.
- The results' presentation in Tables 1 and 2 could be improved for clarity, specifically by explicitly noting the significance of bolded and underlined results in the table captions.

[1] Isensee, F., Jaeger, P. F., Kohl, S. A., Petersen, J., & Maier-Hein, K. H. (2021). nnU-Net: a self-configuring
method for deep learning-based biomedical image segmentation. Nature methods, 18(2), 203-211.
[2] Kromp, F., Bozsaky, E., Rifatbegovic, F. et al. An annotated fluorescence image dataset for training nuclear segmentation methods. Sci Data 7, 262 (2020).

**Detailed Comments:**

- A discussion on the selection of U-Net3+ over nnUnet [1] would enhance the paper's review of the state of the art, given nnUnet's noted performance.
- Clarification on the decision not to explore instance segmentation approaches would be valuable, especially considering the mentioned challenges of overlapping nuclei.
- Discussion on the computational complexity of the proposed approach, in comparison to contenders, is necessary to fully evaluate its practical applicability.
- The manuscript would benefit from a section on the limitations of the proposed work and considerations for future research directions to provide a complete view of the study's context and potential impact.
- Publicly releasing the used source code would be appreciated.

[1] Isensee, F., Jaeger, P. F., Kohl, S. A., Petersen, J., & Maier-Hein, K. H. (2021). nnU-Net: a self-configuring
method for deep learning-based biomedical image segmentation. Nature methods, 18(2), 203-211.

**Justification Of Final Rating:**

The authors have answered all my questions and addressed all concerns. The overall quality of the work was improved. I'm still wandering about the missing comments to the other reviewers concerns. Therefore I updated my final rating to 3a.

**Justification Of The Preliminary Rating:**

The nnUnet, which application in biomedical settings was already shown, is completely missing from the state of the art. Furthermore an explanation on why instance segmentation was not considered is missing.

**Questions To Address In The Rebuttal:**

- Could the authors elaborate on their choice of U-Net3+ over e.g. nnUnet or similar, particularly in relation to their respective performances in similar segmentation tasks?
- What are the reasons behind not considering instance segmentation methods for addressing the challenge of overlapping and clumped nuclei?
- Can statistical evidence be provided to substantiate the claims of significant performance improvements achieved by the proposed model?
- Would the authors be willing to include a combined ablation study that clearly delineates the contribution of each model enhancement to the overall performance?
- How do the authors justify the computational complexity and practical applicability of the enhanced U-Net3+ model compared to other state-of-the-art models?

---

> ### Author Response · Authors · 2024-03-18
>
> Thank you very much for your insightful and constructive feedback on our manuscript. We greatly appreciate the time and effort you have invested in providing a detailed review.
>
> We acknowledge the exceptional performance of nnUnet in various segmentation tasks. Our choice to build upon U-Net3+ was primarily driven by its architectural simplicity, and preliminary experiment results as now detailed in section A. 1. Our preliminary experiment showed that U-Net3+ outperformed nnUNet and ELU-Net and thus we selected it as the base model for potential improvements. Moreover, the simplistic approach was also the reason we went for normal segmentation instead of instance segmentation. We agree that adopting instance segmentation could potentially alleviate some of the inherent problems in nuclei segmentation and increase the accuracy of the model and recognize this as an avenue for future research.
> Upon reevaluation of our manuscript, we found some inaccuracies in Tables 4 and 6 which we have corrected and have also included insets in our figures to enhance their visibility. We have also added experiment results of configuration (32. 64, 128, 256) to table 4 and have included the guided filter experiment result in the ablation section A.7. Furthermore, we also added an ablation overview diagram for better interpretation of the ablation study section and show the improvements in the model performance through the enhancements.
>
> The ablation and almost all parts of the experiments were conducted on MoNuSeg dataset and to justify that the model doesn’t only perform well on MoNuSeg data but performs well on nuclei segmentation, we performed experiments across CPM-17 and CoNSep dataset (Table 2) and the model achieved the state-of-the-art results without any individual optimizations. This demonstrates the ability of the model for nuclei segmentation. Furthermore, we also will be providing the code used for the experiments upon selection for clarity and further research.
>
> To summarize, while there is room for further innovation and refinement in our study, we firmly believe that our research makes a meaningful contribution to the field. By leveraging and optimizing existing architectures, we have advanced the state of the art, underscoring the potential of task-specific models to capitalize on the existing research. Our findings demonstrate that with careful tuning, even established methods can yield superior results.
> We hope that the updates and clarifications provided will satisfy the concerns and render the study a valuable contribution to the field of nuclei segmentation. Thank you for spending valuable time to review our paper and provide meaningful guidance.

---

### Official Review · Reviewer_EpDY · 2024-02-27

**Confidence:** 4
**Preliminary Rating:** 4
**Final Rating:** 5

**Summary:**

The authors describe a model for the common task of nuclei segmentation from pathology images. They have conducted careful experiments on several enhancements to the existing UNet3+ and benchmarked them. The authors arrive upon a combination of enhancements that improve model performance quite significantly over baselines.

**Strengths:**

This paper presents a thorough empirical evaluation of enhancements to baselines for a common task: nuclei segmentation. Enhancements include changing loss function, activation function, preprocessing, augmentation, and architecture. With so many potential enhancements published, I believe this could be a useful paper for readers to see which improvements are actually beneficial for the nuclei segmentation task (and it is likely that this will generalize to similar digital pathology segmentation tasks). The improvement over baselines seems fairly substantial (over 5 percentage points). A thorough ablation study is presented.

**Weaknesses:**

There are no technically novel aspects presented in this paper. All enhancements have previously been published separately, and this paper just brings them together and performs an ablation study. This paper requires a long appendix that contains all the ablation experiments, and  in my opinion the most interesting part of the paper is in the appendix, perhaps suggesting that a MIDL conference paper is not the best place to present this work.

**Detailed Comments:**

- Methods should be cited when they are presented in the methods section (not just the introduction).
- There appear to be some errors in Table 4 (appendix). How can a network with a channel configuration of [16, 32, 64, 128] have more parmeters than one with [32, 32, 64, 128] channels, despite having fewer channels? Other rows of the table have similar problems.
- Similarly there are repeated rows in Table 6. This may be a coincidence, or could be a mistake. However, if the table is correct there is no advantange to to Reinhard-normalization, which contradicts the text.
- There is no need to define standard metrics such as Dice and IoU, these can be left to references. This would save some space to move some information about the ablation experiments from the appendix to the main body.
- There is only one class so mean IoU is the same as standard IoU in this case, no?
- Since SAM requires a prompt, its performance will vary depending  on the prompt. How did the authors address this when calculating a single value for its performance?
- It's not clear whether the model was retrained on the other datasets (CPM-17 and CoNSep) or whether the model trained on MoNuSeg was simply tested on the other datasets.
- Did the authors experiment with alternative in-network normalization methods (batch norm vs instance norm etc etc)
- There appears to be no ablation regarding the trainable guided filter.
- Were there any other potential improvements that the authors made and found did not improve performance and so discarded? It would interesting for readers to know what these were even if they are not fully written up.

**Justification Of Final Rating:**

While a direct response to my comments would have been very useful, it does appear that the authors have addressed all the comments in their revised manuscript (except the one about SAM). I do think the paper is stronger for having included nnUnet (and shown superiority over it). My opinion of the paper has therefore gone up to "strong accept".

Responding to other reviewers' comments: I am rather baffled at the other reviewers' criticism that an ablation study is needed. The entire appendix is a (very thorough) ablation study. Perhaps they simply didn't read the appendix? Though a statistical test would be nice, it is very common not to do this within the ML, rightly or wrongly, so it seems harsh to reject the paper on this basis alone (EDIT: I see from a comment on the thread above that the results are statistically significant according to a t-test, but that this has not made it into the paper). Lack of consideration of instance segmentation, while a valid idea, is rather off topic and not a reason to reject this paper. Semantic segmentation is a common and valid way to perform cell segmentation and a paper considering only semantic segmentation has value. Consequently I quite strongly disagree with reviewer eYtt's criticisms, especially with the nnUnet comparison now in place.

**Justification Of The Preliminary Rating:**

This paper is technically sound. The decision for acceptance comes down to whether a thorough ablation study of various proposals from elsewhere is of sufficient interest to the community.

In my personal opinion, such papers are useful resources that help readers get a sense of which of many potential enhancements are useful and work well in combination. I therefore lean towards accepting this paper, but others may disagree.

**Questions To Address In The Rebuttal:**

No particularly pressing points to address, but see list of questions in the detailed comments.

---

> ### Author Response · Authors · 2024-03-18
>
> Thank you very much for your insightful and constructive feedback on our manuscript. We greatly appreciate the time and effort you have invested in providing a detailed review. Your acknowledgment of the thorough empirical evaluation and the significant improvement our model demonstrates over existing baselines is encouraging.
> We acknowledge your concerns and do agree that there was an oversight on our part in both Tables 4 and 6. These errors were corrected to accurately reflect the parameter counts and the experimental outcomes. We have also added experiment results of configuration (32. 64, 128, 256) to Table 4 and have included the guided filter experiment result in the ablation section A.7. We have further added an ablation overview diagram (Figure 4) for better interpretation of the ablation study section and modified Figure 3, Figure 7 to include insets for better visualization and readability.
> For SAM prediction, we used image masks as prompts for the prediction results. (SAM was not retained but only used for inference)
> The model was trained individually across the datasets and we've also made it explicit in the manuscript as well.
> While we did not experiment with alternative in-network normalization methods in this study, we recognize the value of such investigations and have noted this as a potential direction for future research. Moreover, regarding our choice of baseline for the experiments, our preliminary results indicated that U-Net3+ offered a compelling balance of simplicity and performance, which informed our decision to select it over other architectures like nnU-Net (now presented in A.1.). This simplistic approach was also the reason we went for normal segmentation instead of instance segmentation. However, we accept the value of instance segmentation in enhancing model precision and recognize this as an avenue for future research.
>
> Your constructive criticism and insightful questions have been invaluable in refining our manuscript. We hope these revisions and clarifications address your concerns satisfactorily.  Thank you for spending valuable time to review our paper and provide meaningful guidance.

---

> > ### Author Response · Authors · 2024-03-22
> > **Additional Response to Reviewer EpDY**
> >
> > We have list up the potential questions from the review and have addressed in below.
> >
> > **Question 1:  Are there errors in Table 4 and 6?**
> >
> > Response 1: We extend our apologies for the errors in Tables 4 and 6 and appreciate your understanding. The corrections have been made to accurately display the parameter counts and experiment results. Additionally, we have now included the results for the configuration (32, 64, 128, 256) in Table 4, providing a more comprehensive view of the models' performances.
> >
> > **Question 2: Is mIoU the same as IoU since number of classes is one?**
> >
> > Response 2: Yes, in the context of our binary segmentation task, where nuclei regions are labeled as 1 and non-nuclei regions as 0, the mean Intersection over Union (mIoU) is effectively equivalent to the standard IoU.
> >
> > **Question 3: How did inference using SAM was performed?**
> >
> > Response 3: For SAM, we utilized image masks as input prompts to guide the inference process. It's worth noting that SAM was not re-trained but was used solely for inference purposes.
> >
> > **Question 4: Whether model was trained on other datasets ?**
> >
> > Response 4: Yes, our model was individually trained on each dataset, and the performance metrics were computed using their respective test sets. This has now been clearly stated in Section 3 of our manuscript.
> >
> > **Question 5: Did any other in-network normalization was performed?**
> >
> > Response 5: We did not experiment with alternative in-network normalization methods in this study, but we recognize the value of such investigations and have noted this as a potential direction for future research.
> >
> > **Question 6: Why is there absence of ablation on trainable guided filter?**
> >
> > Response 6: We have included the trainable guided filter section in the ablation studies under section - A.7. We have also added a comparative qualitative visualization of effects of the guided filter in Figure 8.
> >
> > We hope these adjustments and clarifications address your concerns and we look forward to any further comments or suggestions

---

### Official Review · Reviewer_eYtt · 2024-02-28

**Confidence:** 4
**Preliminary Rating:** 1
**Final Rating:** 3.5

**Summary:**

The authors provide a study to segment cell nuclei in histopathology images. In particular, they mine the U-Net architecture and combine several well-established techniques to outperform baselines, such as U-Net, U-Net++ and SAM. This fact relies on a couple of architectural designs, such as Gated Linear Units (GLUs), improved skip connects (inherited from U-Net++), and state-of-the-art loss combinations (Focal and Dice loss).

**Strengths:**

The paper is straight forward written, the appendix contains a lot of more detailed information about the design choices, and the comparison is made on three independent datasets and across several baselines. It seems that three choices are essential: loss function, activation function and regularization (dropout strategies).

**Weaknesses:**

In general, I would actually like to see the author's final model, and then do an ablation strategy. It seems like the design choices are independent, and then the best of all worlds is taken. However, I believe the authors can make stronger claims on their design choice if they do provide their model, ablate or change the design choice (e.g. from GLU back to ReLU, Dice Loss), and then show clearly, that only their combination is strong. This would give very much insight into the complex field of architectural design strategies.

Further, claims such as "especially in conjunction with Max Blur Pooling" are very strong, despite the fact that adding MBP only has marginal benefits (Appendix A.3). I recommend toning down a lot of the statements and adjust them according to the authors' findings. Similar is for example the stain normalization. Table 6 shows no improvement upon stain normalization.

This problem (nuclei segmentation) is not only semantic, but also related to instance segmentation. I am missing the context if the authors model would bring benefits in this regard.

Finally, I am missing very strong baselines. For example, nnU-Net has been shown several times, that the original U-Net is very strong. For example, using the same loss and GLUs in the normal U-Net and compare it to nnU-Net would largely improve the statements of the paper.

**Detailed Comments:**

* I would largely check on the papers' content and highlight the relevant findings towards the architecture
* There are some spelling errors, e.g. implmentation on page 8.
* Some Figures would benefit from insets, such as Figure 3, to see clearly the performance.

**Justification Of Final Rating:**

I don't know if the appendix was as strong as before, but I believe that the authors did a good job in incorporating the other reviewers' comments and updated the paper accordingly, as well as the incorporation of the nnUnet baseline. I think I was a little harsh on the authors and apologize.

**Justification Of The Preliminary Rating:**

I believe in the current form, the manuscript has two very strong flaws (no real ablation study and missing of strong baselines), that I do not believe that the authors can cope with the demands in the short rebuttal time. The study is of general interests, but lacks useful experiments and statements in the current form.

**Questions To Address In The Rebuttal:**

* Ablation study and real impact of their design choices
* Strong baselines
* and commenting on the remaining comments

---

> ### Author Response · Authors · 2024-03-18
>
> Thank you sincerely for your thoughtful and thorough review of our manuscript. Your expertise and detailed analysis have provided invaluable insights to our research. Your detailed feedback is invaluable, and we're particularly encouraged by your appreciation of the manuscript's clarity.
>
> We acknowledge your concerns about the necessity of ablation study for the validation of our architectural decisions. Indeed, our approach of comparing incremental model modifications to a baseline did not fully exploit the required potential. While our current experiments reveal the impact of our design choices, they fall short of illustrating their independent effectiveness.  We should have fixed a final model and then ablate to find the presence and impact of certain design choices ( such as GLU, Loss, etc. ), but we performed experiments where we compared the results with the baseline. Nevertheless, these exploratory analyses did reveal the influence of each design choice on our model's performance. To elucidate these findings, we have now included an overview diagram of the ablation study (Figure 4)  in the revised manuscript, which clarifies the contribution of each enhancement to the overall effectiveness of our approach.
>
> We have taken care to address the overstatements in our initial manuscript and ensure that our conclusions are firmly grounded in the data we have presented. Upon reevaluation of our manuscript, we found some inaccuracies in Table 4 and 6 which we have corrected and have also included insets in our figures (Figure 3, Figure 7) to enhance their visibility.
>
> Regarding our choice of baseline for the experiments, our preliminary results indicated that U-Net3+ offered a compelling balance of simplicity and performance, which informed our decision to select it over other architectures like nnU-Net (now presented in A.1.). This simplistic approach was also the reason we went for normal segmentation instead of instance segmentation. However, we accept the value of instance segmentation in enhancing model precision and recognize this as an avenue for future research.
>
> To summarize, while there is room for further innovation and refinement in our study, we firmly believe that our research makes a meaningful contribution to the field. By leveraging and optimizing existing architectures, we have advanced the state of the art, underscoring the potential of task-specific models to capitalize on the existing research. Our findings demonstrate that with careful tuning, even established methods can yield superior results.
> We hope that the updates and clarifications provided will satisfy the concerns and render the study a valuable contribution to the field of nuclei segmentation. Thank you for spending valuable time to review our paper and provide meaningful guidance.

---

> > ### Author Response · Authors · 2024-03-22
> > **Additional Response to Reviewer eYtt**
> >
> > We have list up the potential questions from the review and have addressed in below.
> >
> > **Question 1: What are the impacts of design choices and ablation study?**
> >
> > Response 1: The ablation study was structured to sequentially assess the impact of our design choices. While it doesn’t quantify the exact extent of each choice's contribution, it does clearly demonstrate their effectiveness. By following a sequential ablation approach and observing consistent improvements in model accuracy, we establish that each modification indeed contributes positively towards achieving state-of-the-art results. This is effectively communicated through the overview diagram in Figure 4, where the incremental benefits of our design choices are made evident.
> >
> > **Question 2:  How are the baselines selected?**
> >
> > Response 2: In identifying a strong baseline for our study, we prioritized simplicity and efficacy in managing both local and long-range dependencies within the histopathological  context. We selected three state-of-the-art models—U-Net3+, ELU-Net, and nnUNet—due to their demonstrated capabilities. Each model was trained on our dataset without  modifications, and their performances were tabulated in Table 3 of the ablation study. U-Net3+ emerged as the most suitable baseline for further enhancements based on its initial performance, setting the foundation for our subsequent optimizations.
> >
> > **Question 3: There are some strong claims in manuscript despite having marginal benefits. Is it possible to tone down the statements based on findings?**
> >
> > Response 3: Acknowledging the reviewer's feedback, we have carefully reviewed and revised our manuscript to ensure that all claims are accurately represented and fully supported by our data. We addressed discrepancies in Table 4 and Table 6 concerning feature channel configurations and stain normalization effects. These tables have been updated to correctly reflect the outcomes of our experiments, thus ensuring our statements accurately mirror the empirical evidence.
> >
> > **Question 4: Is it possible to insert insets for performance visualization?**
> >
> > Response 4: In response to this valuable suggestion, we have updated Figures 3 and 8 to include insets. These insets offer a clearer and more detailed visualization of performance enhancements of the model.
> >
> > **Question 5: Why there is no comparison of design choices on other models (especially original U-Net) for better evaluation ?**
> >
> > Response 5: We acknowledge the importance of comparing our design choices across different networks, such as the original U-Net, to comprehensively evaluate their effectiveness. While our current study focused on achieving SOTA results using a lightweight model architecture by selectively incorporating a few promising components tailored specifically nuclei segmentation task. We recognize the potential of extending our proposed enhancements to other models.
> >
> > We hope these adjustments and clarifications address your concerns and we look forward to any further comments or suggestions

---

### Comment · Area_Chair_MTVN · 2024-03-19
**paper is open for discussions**

Dear Reviewers The authors have submitted their rebuttal addressing the raised questions. The paper remains open for further discussion and engagement.

---

> ### Comment · Reviewer_rzgC · 2024-03-21
>
> Sorry but it is very hard to consider this rebuttal, as the authors neither responded to the raised questions and concerns here nor highlighted the changes in their revised document.
>
> Can we expect some comments from the authors?
>
> Thank you!

---

> > ### Author Response · Authors · 2024-03-22
> > **Additional Reponse to Reviewer rzgC**
> >
> > We would like to extend our sincerest apologies for any oversight in directly addressing the questions and concerns. We have listed them below with our remarks.
> > **Question 1: Could the authors elaborate on their choice of U-Net3+ over e.g. nnUnet or similar, particularly in relation to their respective performances in similar segmentation tasks?**
> >
> > Response 1 Our preliminary experiments indicated that U-Net3+ achieved superior Dice scores over nnUNet and ELU-Net, leading to its selection as our baseline model. In this rebuttal we updated the results of nnUNet in Table 3.
> >
> > **Question 2: What are the reasons behind not considering instance segmentation methods for addressing the challenge of overlapping and clumped nuclei?**
> >
> > Response 2: We acknowledge the significance of instance segmentation in addressing the challenges in nuclei segmentation. However, our study focused on improving the existing segmentation model by applying various techniques and observing the presence of their effectiveness. There is still a surge in active research for nuclei binary segmentation and our study could help potential future research to opting for certain design choices in their model architecture.
> >
> > **Question 3: Can statistical evidence be provided to substantiate the claims of significant performance improvements achieved by the proposed model?**
> >
> > Response 3: Indeed, the focus of our research is to present that SOTA results are attainable using a lightweight model architecture by selectively incorporating a few promising components tailored specifically for the task of nuclei segmentation. Our ablation study does shed light on the effectiveness of our design choices, showing incremental improvements in model performance, finally leading to over six percent increase over the base model. Moreover, we perform experiments with the same model and same hyperparameter values on other datasets - CPM-17 and CoNSep and the proposed model shows performance gains in terms of Dice scores.
> >
> > **Question 4: Would the authors be willing to include a combined ablation study that clearly delineates the contribution of each model enhancement to the overall performance?**
> >
> > Response 4: Indeed, we have added an overview of the ablation study which breaks it into three enhancements as shown in Figure 4 where the enhancements sequentially raised the Dice score from 0.826 to 0.8902.
> >
> > **Question 5: How do the authors justify the computational complexity and practical applicability of the enhanced U-Net3+ model compared to other state-of-the-art models?**
> >
> > Response 5: Our experiments show that the proposed design choices along with other enhancements increase the overall performance of the baseline model and achieves the SOTA performance. The enhanced U-Net3+ model, while not using pre-trained weights and being trained from scratch with limited data, is able to perform considerably better. While more experiments are still required to ascertain the claims of the design choices of our model, our study does show potential improvements and practical applicability, especially in settings with limited resources.
> >
> > We hope that the clarifications address your concerns.We thank you once again for your patience and guidance and look forward to any further comments or suggestions you may have.

---

> ### Comment · Reviewer_rzgC · 2024-03-22
>
> Thank you for providing these comments to my questions!
>
> I guess my fellow reviewers would like to have their questions answered too.
>
> The addition of the nnunet results is appreciated. It would be great, if you would also add it to the state of the art section of you work.
>
> Furthermore I still have an issue with claiming significant improvements without a statistical proof.

---

> ### Author Response · Authors · 2024-03-23
> **Adiitional Response**
>
> We greatly appreciate your further comments.
>
> Regarding your concerns about the claim of significant improvements without statistical proof, we performed a two-sample t-test  to compare the Dice scores of our proposed model against the base model (U-Net3+). The analysis yielded a p-value of approximately 0.0134. The p-value is less than the conventional threshold of 0.05, indicating that there is a statistically significant difference between the mean Dice scores of the two models. This result suggests that the performance difference observed between our proposed model and the base model is unlikely to have occurred by chance, supporting the conclusion that our model has a statistically significant improvement in Dice scores compared to the base model. ​
>
> We have made the following changes to the manuscript.
> 1. We have now included the nnU-Net results in the state-of-the-art section of our work, as suggested.
> 2. We have added the result of two-sample t-test in the section 3 - Experiments and Results.
> 3. We removed the equations for the evaluation metrics while citing them in references (as suggested by Reviewer EpDY ) to adhere to the page limit.

---

### Meta-Review · Area_Chair_MTVN · 2024-04-03

**Recommendation:** Accept (Poster)
**Confidence:** 5

**Metareview:**

Two of the three reviewers support accepting this work as a poster and believe it will generate meaningful discussions. I believe the rebuttal has addressed most of the important comments. The authors are requested to revise the camera-ready version in accordance with the feedback provided in the rebuttal.

---

### Decision · Program_Chairs · 2024-04-05

Accept (Poster)